# Towards routine chromosome-scale haplotype-resolved reconstruction in cancer genomics

Shilpa Garg[1,2] ✉

Cancer genomes are highly complex and heterogeneous. The standard short-read sequencing and analytical methods are unable to provide the complete and precise base-level structural variant landscape of cancer genomes. In this work, we apply high-resolution long accurate HiFi and long-range Hi-C sequencing to the melanoma COLO829 cancer line. Also, we develop an efficient graph-based approach that processes these data types for chromosome-scale haplotype-resolved reconstruction to characterise the cancer precise structural variant landscape. Our method produces high-quality phased scaffolds on the chromosome level on three healthy samples and the COLO829 cancer line in less than half a day even in the absence of trio information, outperforming existing state-of-the-art methods. In the COLO829 cancer cell line, here we show that our method identifies and characterises precise somatic structural variant calls in important repeat elements that were missed in short-read-based call sets. Our method also finds the precise chromosome-level structural variant (germline and somatic) landscape with 19,956 insertions, 14,846 deletions, 421 duplications, 52 inversions and 498 translocations at the base resolution. Our simple pstools approach should facilitate better personalised diagnosis and disease management, including predicting therapeutic responses.

Cancer is a genetic disease resulting from the accumulation of mutations in genes that regulate cell division, survival, invasion or other hallmarks of the transformed phenotype[1]. To identify and characterise the mutations, a chromosome-scale haplotype-resolved genome reconstruction approach is important. The mutation events include chromosomal insertions, deletions, duplications, translocations, inversions, which range in size from 50 bp to whole chromosomes[2]. There are also thousands of rearrangements (structural variant - SVs), occurring as a single event on the same chromosome or across different chromosomes. There has been a plethora of studies, including many consortium studies, such as the ICGC-TCGA Pan-cancer analysis of whole genomes (PCAWG), reporting the occurrence of these SV events using short-read WGS data[2,3]. However, the high-resolution

methods are scanty although these have an added advantage (explored in this paper) for comprehensive structural variation landscape especially using long accurate HiFi and long-range Hi-C technologies. Also, the high-resolution tools have an advantage over short-read studies in phasing mutations (i.e., which allele carries the mutation) on the chromosome level that is specifically important for studying the clonal evolution and progression of cancer, especially when one allele is the wild-type and the other is a mutant[4].

Importantly, the high-resolution PacBio HiFi sequencing method produces reads longer than 20 kb with error rates of ~1%. The chromosome conformation capture Hi-C sequencing method produces short-reads connecting loci multi-megabase apart on the whole-chromosome level[5]. These high-resolution sequencing techniques

[1]NNF Center for Biosustainability, Technical University of Denmark, Kongens Lyngby DK-2800, Denmark. [2]Department of Biology, University of Copenhagen, Copenhagen DK-2200, Denmark. ✉e-mail: sgarg@biosustain.dtu.dk

are of utmost importance, however, not applied for cancer genomes. The most widely analytical tools for processing data types are HiCanu[6], hifiasm[7], and Flye;[8] however, they are not designed for cancer genomes for studying chromosomal-level events and the alleles (haplotypes) in which the SV events occur. Indeed, the recent genome reconstruction approach reported by Cheng et al.[7] uses trio information that further limits its applications because trios are not routinely available. A recent publication (https://arxiv.org/abs/2109.04785v1) presented an Hi-C extension of hifiasm for phased contigging for non-cancer studies, but has limitations to produce chromosome-level haplotype-resolved assemblies necessary for comprehensive SV landscape in personalised cancer genomes (without reference bias). Further, several studies have sought to follow the scaffolding approach, such as 3D-DNA[9] and Salsa2[10], which can detect chromosome-scale events, but these techniques have the underlying assumption of haploidy and cannot produce haplotype-aware scaffolds that are specifically important in terms of chromosome aneuploidy events. Although the latest scaffolders Falcon-Phase[11] and AllHi-C[12] restore phasing information, they are only designed for long erroneous reads and Hi-C, resulting in genomes of poor continuity and accuracy. While our recent WHdenovo[13] method is the first to use phasing during the assembly process designed only for small genomes, the recent Dipasm method[14] for human-scale-size genomes loses haplotype information in complex, structurally rearranged regions and is not suitable for phased scaffolding of cancer genomes. Thus, no techniques are available for cancer genomes that combine phasing and scaffolding/assembly on chromosome level, and are suitable for use in routine clinical high-quality genome production. In this work, we develop an efficient graph-based approach that processes high-resolution HiFi and Hi-C data types for chromosome-scale haplotype-resolved reconstruction to characterise the cancer precise structural variant landscape. Our approach finds the precise comprehensive structural variant (germline and somatic) landscape with 19,956 insertions, 14,846 deletions, 421 duplications, 52 inversions, and 498 translocations at the base resolution.

## Results

### Sequencing and computational method for COLO829

We sequenced the melanoma COLO829 cancer cell line ((https://www.lgcstandards-atcc.org/products/all/CRL-1974.aspx/?geo_country=nl—matched tumour-normal) with long accurate HiFi and long-range Hi-C sequencing (with the necessary support of technologists – see "Methods") to produce an open resource for the community, as well as to advance the characterisation of cancerous mutations at the base and haplotype resolution (see Fig. 1). In addition, we developed an efficient graph-based computational tool to process these data types in the context of cancer genomes that can consider inter- and intra-chromosomal structural genome complexity, along with haplotype information, in an integrative graph algorithm. Our graph approach provides a compact representation to combine multiple data types in a joint sequence space that is useful for preserving any levels of genome complexity to produce complete phased genomes directly. The graph can also effectively characterise SVs in repetitive regions (see Fig. 1) and can perform polyploid phasing in the presence of aneuploidy and transposable elements at the whole-chromosome level. For example, we applied our graph-based method for somatic and germline calling for COLO829, where somatic calls were produced by subtracting the germline calls for COLO829BL[15,16] and HG002 GIAB benchmark (https://ftp-trace.ncbi.nlm.nih.gov/ReferenceSamples/giab/release/AshkenazimTrio/HG002_NA24385_son/latest/GRCh37/). As a result, we found 166 somatic structural variant calls at precise base-level resolution on the whole genome. We even specifically characterised these calls in complex repeat elements including ALUs, L1, L2, and benchmarked against publicly available calls produced using multiple technologies/tools, for example, a short-read-based NYGC call set (https://www.nygenome.org/bioinformatics/3-cancer-cell-lines-on-2-

sequencers/), single-cell-based call set (available from Enrique Velazquez-Villarreal et al. 2020)[17] and multi-technology-based UMCU call set (https://github.com/UMCUGenetics/COLO829_somaticSV). Figure 1b presents the 4-way comparison of these callsets, where each bar represents the agreement of callsets in repeat elements. We observed that our calls agreed with the single-cell benchmarked callset (blue) and UMCU (yellow), but our method provides more precise SV characterisation at the base and haplotype resolution. The green and red and pink coloured bar represent calls missed by pstools possibly due to low variant allele frequency but found in single-cell method. Additionally, we produced higher calls in comparison to bulk short-read-based calls from NYGC. Out of these somatic variants, we found 15 calls and 6 calls that fall in the COSMIC genes. Figure 1c presents the distribution of germline SV calls in repeat elements, indicating the higher rate of germline calls than somatic calls in Fig. 1b. From above, it follows that our pstools method can produce precise and comprehensive haplotype-aware SV landscapes over short-read sequencing technologies in repetitive regions.

Next, Fig. 2 presents the workflow of the whole algorithm in detail. Based on our previous Sdip algorithm[18], the pstools algorithm initially generates a sequence graph that preserves the haplotype information in the overlapping HiFi read sequences using hifiasm on the whole-genome level. The resulting sequence graph contains multiple components that represent either the same or different chromosomes. Afterwards, from every component, pstools detects specific structures —simple/complex bubbles and branches—that represent heterozygous sites and/or structural rearrangements. Then, it maps the Hi-C reads through the HiFi sequence graph that provides global phasing, ordering and orientation information to produce haplotype paths from a series of bubbles/complex structures. Unlike other scaffolding methods, the mapping step in pstools is fast and uses phasing information on the graph space. Based on Hi-C mapped reads, it first produces haplotype sequences through bubbles in every component (phased contigs) and then connects these phased contigs in the correct ordering with proper orientation. The process of connecting haplotype sequences from components in their correct ordering and orientation is called haplotype-aware scaffolding. This task is challenging due to the explosion in combinatorial search space needed to connect pairs of phased contigs while also considering phasing information in this process. The Methods section describes how the pstools algorithm connects phased contigs by the neighbourhood property of contigs, such as adjacent contigs, and has comparable read support. The whole workflow is very fast, as it can be completed in less than 12 h. Our pstools algorithm combines the advantages of HiFi and Hi-C in an efficient manner to produce accurate, continuous and complete haplotypes of complex cancer genomes for routine clinical applications. The chromosome-scale haplotype-resolved genome assemblies are used to call SVs using SVIM-asm[19]. Fig 3 shows the advantages of the pstools algorithm: (1) it correctly disentangles the chromosomes (not resolve repeats, but connect components using long-range Hi-C with 100 N's), as demonstrated for chr13 and chr14, and (2) it connects the arms of chromosomes correctly, as demonstrated for chr1 and chr6, making it useful for inter- and intra-chromosomal structural sequence events in routine clinical applications.

### Benchmarking

We initially benchmarked the pstools algorithm on healthy human samples (HG002, HG00733, and PGP1)[14] that are publicly available and are used in many assembly studies. For benchmarking, we obtained recent state-of-the-art trio data: trio-hifiasm[7] contigs and/or ran salsa2 for scaffolding. We computed standard evaluation metrics: NG50 for continuity, switch/hamming errors for phasing evaluation at the whole-chromosome level, and total sequence length. Table 1 presents the statistics on the evaluation metrics of phased sequences, computed on gapped assembly. In all human experiments, the G value is

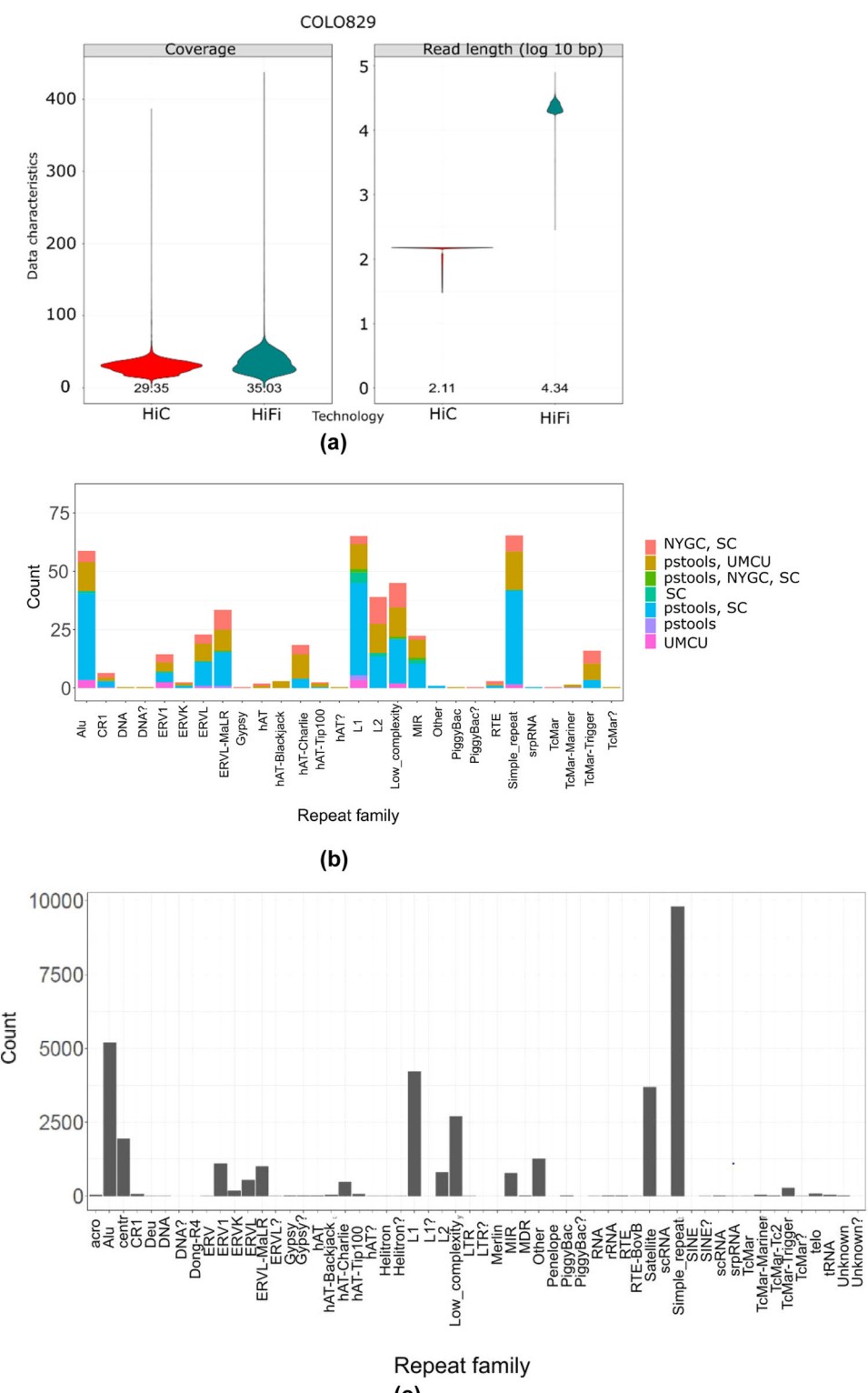

**Fig. 1 | COLO829 HiFi/Hi-C sequencing and SV discovery. a** Read length and coverage characteristics of HiFi and Hi-C sequencing of COLO829 (Top). **b** Benchmarking of somatic SVs in repeat elements (Middle). SV call sets: short-read-based NYGC call set (https://www.nygenome.org/bioinformatics/3-cancer-cell-lines-on-2-sequencers/), single-cell-based call set available from Enrique Velazquez-Villarreal et al. 2020[17], multi-technology-based UMCU call set (https://github.com/UMCUGenetics/COLO829_somaticSV) and pstools. Each bar shows the number of variants agreed between call sets for specific repeat elements. **c** Germline SV calls in repeat elements. *X*-axis: repeat elements, *Y*-axis: number of variants (Bottom). Source data are provided as a Source Data file.

6.0 Gb for NG50 assembly size calculations. Interestingly, the pstools algorithm produces high-quality assembly with a scaffold size of >6.0 Gb, a NG50 assembly size of >130 Mb. In contrast, the competing hifiasm (hi-c) method produces an assembly with a NG50 assembly size

of <52 Mb, indicating that it is not designed for chromosome level genomics. The trio version of the same method (trio-hifiasm) that uses paternal information produces assembly with a NG50 assembly size of <79 Mb in contrast to assembly size of >130 Mb from our method,

## Integration of Hifi and Hi-C in the graph space

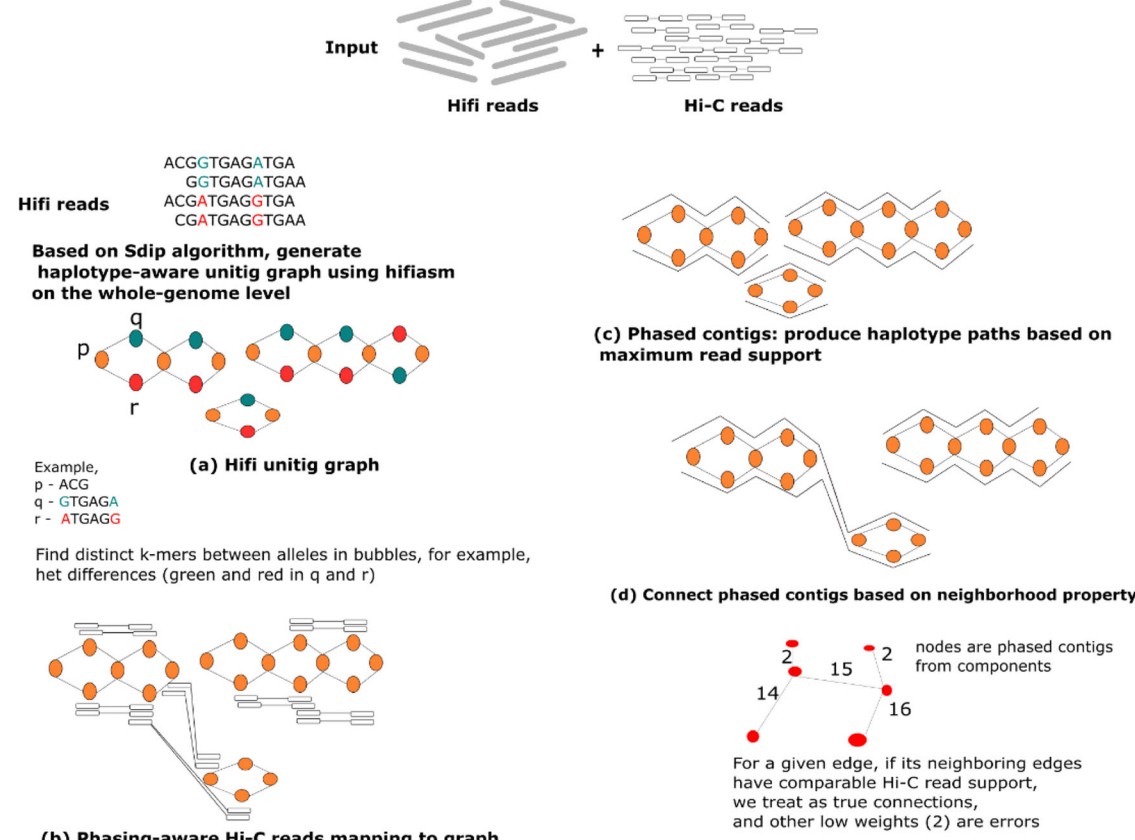

**Fig. 2 | Overview of the pstools algorithm. a** Produce a HiFi sequence graph that retains bubbles and any complex events, **b** Map the Hi-C reads to node sequences in the sequence graph, **c** Phase the bubble chains in the graph to produce haplotype paths (phased contigs), **d** Connect haplotype paths across components to produce phased scaffolds.

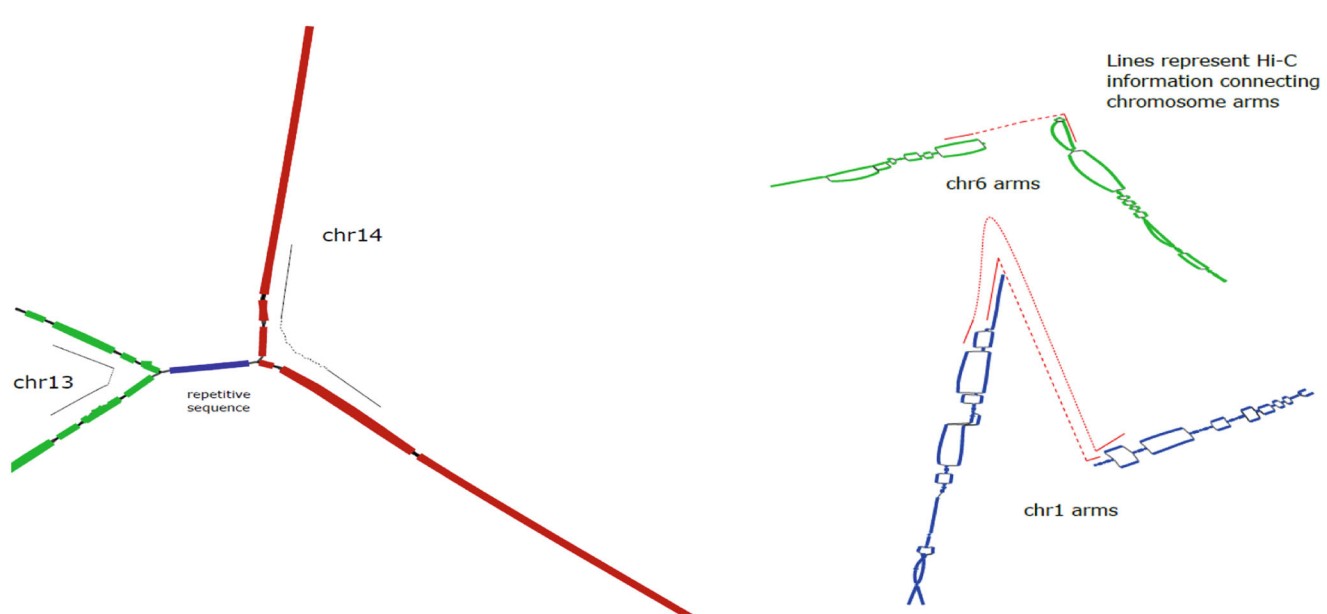

**Fig. 3 | Left: Regions from two chromosomes (chr13 and chr14) are fused in a component due to a common repetitive sequence in HG002.** The Hi-C information in the graph (specifically along each green and red path) is helpful for disentangling the chromosomes. Right: The starting regions of chromosome (chr6 and chr1) arms occur in different components. The phasing information for Hi-C (connecting alleles in bubbles) is useful for accurately connecting the starting regions of chromosome arms.

**Table 1 | Evaluation statistics of phased sequences**

| Dataset | Method | Total size | NG50 | NGA50 | QV | Hamming/switch | Total time | Scaffolding time | Peak RAM (GB) |
|---------|--------|-----------|------|-------|-----|----------------|-----------|------------------|---------------|
| HG002 | pstools[S] | 6.13 Gb | 132.3 Mb | 81.9 Mb | 51.6 | 1.2/0.4 | 12.6 h | 5.1 h | 149 |
| | trio-hifiasm[C] | 5.94 Gb | 78.9 Mb | 64.1 Mb | 51.5 | 1.6/0.8 | 8.1 h | | 135 |
| | Hifiasm (hi-c)[C] | 5.92 Gb | 53.7 Mb | 33.7 Mb | 52.0 | 0.9/0.9 | 10.8 h | | 150 |
| | Dipasm (3D-DNA)[C] | 5.93 Gb | 24.7 Mb | 13.9 Mb | 41.2 | 0.5/0.5 | 252 h | 239 h | 512 |
| | pstools+salsa2[S] | 6.13 Gb | 125.5 Mb | 70.7 Mb | 51.5 | 49.5/0.72 | 27.1 h | 15.6 h | 155 |
| | trio-hifiasm + salsa2[S] | 5.96 Gb | 137 Mb | 70.9 Mb | 51.5 | 1.1/0.6 | 25.8 h | 17.7 h | 167 |
| HG00733 | pstools[S] | 6.21 Gb | 131.9 Mb | 70.7 Mb | 51.1 | 0.6/0.2 | 12.4 h | 4.0 h | 135 |
| | trio-hifiasm[C] | 6.17 Gb | 43 Mb | 55.3 Mb | 51.1 | 1.2/1.8 | 7.1 h | | 139 |
| | Hifiasm (hi-c)[C] | 6.06 Gb | 42.5 Mb | 29.9 Mb | 50.0 | 1.3/1.0 | 9.1 h | | 147 |
| | Dipasm[C] (3D-DNA) | 5.93 Gb | 25.7 Mb | 16.3 Mb | 41.3 | 1.2/0.3 | 271 h | 257 h | 512 |
| | pstools+salsa2[S] | 6.17 Gb | 155 Mb | 70.5 Mb | 51.1 | 48.5/0.52 | 25.0 h | 14.6 h | 160 |
| | trio-hifiasm + salsa2[S] | 6.07 Gb | 135.5 Mb | 70.4 Mb | 51.1 | 1.1/0.9 | 24.0 h | 16.9 h | 161 |
| PGP1 | pstools[S] | 6.14 Gb | 111.7 Mb | | - | 1.2/0.9 | 10.8 h | 3.9 h | 138 |
| | pstools+salsa2[S] | 6.13 Gb | 120.5 Mb | 69.7 Mb | - | 48.5/0.8 | 26.4 h | 16.6 h | 134 |
| | Dipasm[C] (3D-DNA) | 5.94 Gb | 16.5 Mb | 10.65 Mb | - | 1.7/0.3 | 255 h | 242 h | 512 |
| COLO829 | pstools[S] | 6.13 Gb | 132.1 Mb | 54.8 Mb | - | 1.3/0.6 | 10.0 h | 4.1 h | 135 |
| | pstools+salsa2[S] | 6.12 Gb | 100.1 Mb | 54.7 Mb | - | 47.5/0.3 | 26.2 h | 17.1 h | 156 |
| | Hifiasm+salsa2[S] | 6.03 Gb | 1.7 Mb | 0.6 Mb | - | - | 23.1 h | 15.6 h | 145 |
| | Hifiasm +3D-DNA[S] | 6.05 Gb | 2.5 Mb | 1.2 Mb | - | - | 251.5 h | 244 h | 512 |
| | Dipasm[C] (3D-DNA) | 5.92 Gb | 12.4 Mb | 5.6 Mb | - | 20.5/5.6 | 254 h | 239 h | 512 |

NG50/NGA50 is defined as the sequence length/alignments of the shortest contig at 50% of the total genome size (Quast); QV is phred scaled per-base quality (yak); switch errors are local haplotypic switches (yak (a)/GIAB (b)/pstools (c)), while hamming errors are on global scale that measures haplotypic mis-assemblies. Right three columns: Computational resource (time and memory). Scaffolding time includes Hi-C mapping and subsequent steps. Tools are run in the default setting: for salsa2, the Hi-C analysis is run for each haplotype separately. Superscripts in column 2 represent contigs (C) or scaffolds (S). Total time: all steps from raw sequencing data to finish.

suggesting trio information alone can't produce chromosome-scale phased genomes due to unresolved complex repeats. Also, when we performed a salsa2 Hi-C scaffolding step for trio-hifiasm assemblies, we observed that the assembly quality performance of our method is comparable to trio-hifiasm+salsa2 in terms of a phasing accuracy of >98.5% against Genome in a Bottle (GIAB) variant calls (see Table 1) for HG002 as per availability. In addition, our method is faster in runtime (less than 12 h) than the competing triohifiasm+salsa2 method that took much longer (>2 days). Also, the latter has limited applications due to its dependency on trio information. Further, in single-individual (non-trio) experiments, the high hamming error rate in pstools +salsa2 suggests that salsa2 is not designed to incorporate phasing in the scaffolding step. Finally, with one phased segment, the switch error rate and hamming error rate in the clinically important HLA region are <0.01 and <0.03% respectively as against GIAB variant calls.

To further confirm any mis-assemblies, we produced ideograms of alignments of phased sequences from our pstools method to the reference sequence for every chromosome. We have used Grch38 as the reference sequence and the alignment operation is performed using minimap2 (minimap2–paf-no-hit -a -x asm5-cs -r2k -t 20 <ref.fa > <hap.fa >). In Fig. 4, there is one colour for every chromosome, indicating the production of chromosome level haplotype sequences. Further, we used a curation tool (https://gitlab.com/wtsi-grit/rapid-curation) for scaffolding evaluation. We observed that the contigs were grouped into chromosomes correctly. Of 304 contigs, 141 were grouped into 24 scaffolds and the remaining were shorter belonging to repeats (~11 Mb total in length). Of these 141, there were 35 misjoins, 2 missed joins and 0.02% estimated false duplication sequence. For the competing triohifiasm+salsa2 method, there were

65 misjoins, 0 missed joins and 0.42% estimated false duplication sequence. This suggests that our method produces high-quality phased scaffolds compared to triohifiasm+salsa2. Also, we visually checked our assemblies using Hi-C maps (from juicer) that further confirmed the assembly quality. From this, it follows that our suggested pstools facilitates the best high-quality chromosome-scale phased sequences for a single individual, and it does not require any trio information.

Further, we benchmarked our pstools method on the COLO829 cancer line (see Table 1). For comparison, we performed an independent experiment by using the state-of-the-art HiFi contigger (Hifiasm) and the Hi-C scaffolder (salsa2). As expected, the sequence continuity NG50 is very low, confirming that Hifiasm+salsa2 are not designed to reliably reconstruct phased genomes on the whole-chromosome level. We performed an additional experiment with hifiasm and 3D-DNA, where the performance is again poor with a low sequence continuity (NG50 < 2.5 Mb) that consumes high computational resources (time >240 h). Computation of comparable parameters of trio-hifiasm requires additional information, that is trio data for the COLO829 sample, which is not available. Therefore, we skipped trio-hifiasm for comparison. In contrast, our pstools produces high-quality assemblies at the chromosome level in terms of completeness, accuracy and continuity (NG50 > 130 Mb) for the cancer sample and does not require any trio information.

To further assess the structural accuracy of COLO829, we applied WHdenovo (whdenovo simulate: (https://github.com/shilpagarg/WHdenovo/tree/master/whdenovo) to simulate haplotypes by incorporating COLO829 ground truth SNP and SV mutations in the Grch37 human genome. The HiFi and Hi-C read simulations are performed at

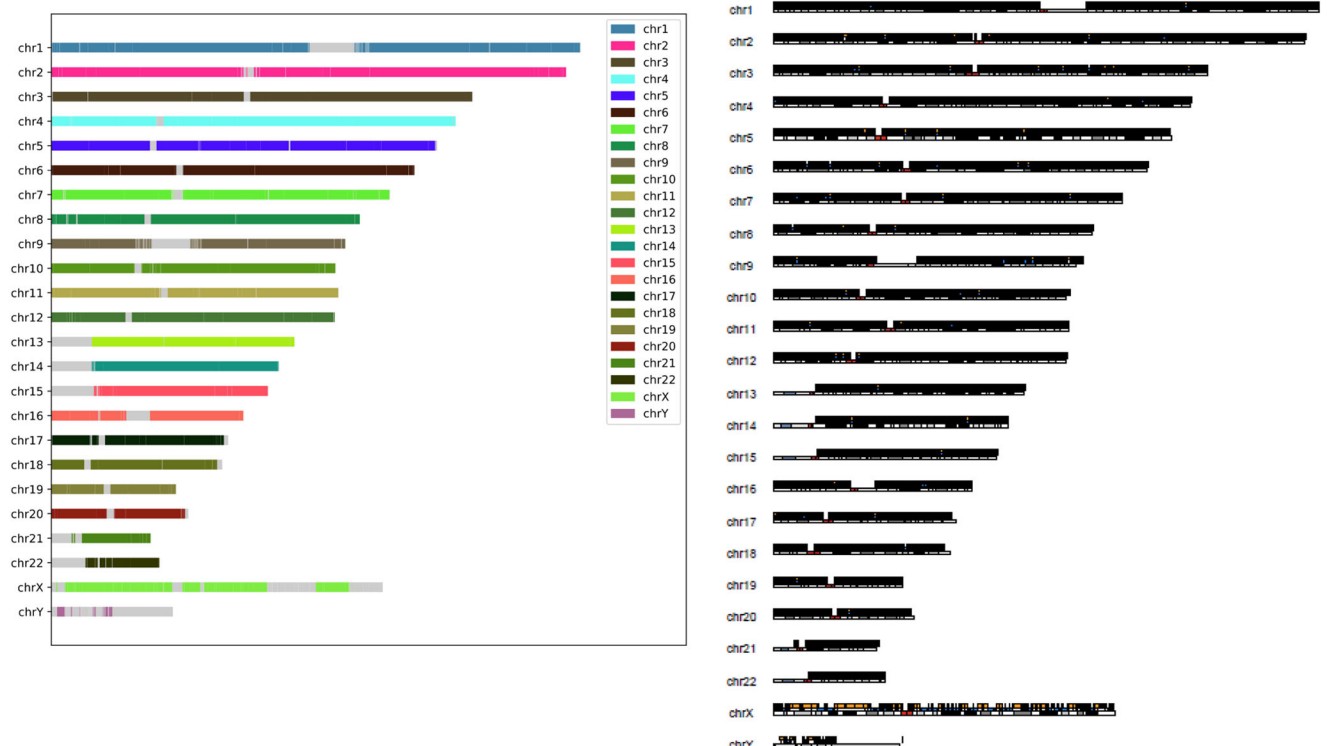

**Fig. 4 | Ideogram of phased sequences of HG002 (left) and COLO829 (right).** One colour for each chromosome representing chromosome level. No colour represents gaps for complex regions, for example, centromeric, acrocentric, etc.

40× fold using pbsim and wgsim respectively. We then applied pstools on these data types that produced assemblies with a NG50 scaffold size of 131.5 Mb. We also additionally compared predicted SV calls against the ground truth and the resultant precision and recall are 97.1 and 95.8% respectively. These results suggest that the pstools produces high-quality assemblies and SV calls. Thus these simulation experiments further confirm the structural accuracy of pstools assemblies.

The run time of our method for human genomes, including initial graph construction and subsequent chromosome-scale genome reconstruction steps, is less than 12 h, while the competing methods, including the salsa2 method, take a longer time, about a few days. The peak memory usage by our method is less than 150 Gb. Table 1 summarises the computational resources in terms of time and memory usage for each dataset, suggesting that our newly developed pstools algorithm is significantly faster (by an order of magnitude) and enables routine production of high-quality haplotype-resolved genomes.

**Whole-genome precise SV characterisation of COLO829**

Further, we aimed to identify and characterise the precise haplotype-resolved structural variations (germline and somatic) of the complex cancer genome[15,16]. In total, we found 19,956 insertions, 14,846 deletions, 421 duplications, 52 inversions, and 498 translocations at base resolution (see Fig. 5). We assessed the quality of our call set by initial comparison against multiple technologies because there is no ground truth publicly available[20]. We first prepared SV benchmark calls (germline and somatic) from multiple technologies (PacBio CLR, Nanopore, PacBio HiFi and short-read sequencing) and ran standard tools for calling SVs for the respective data types (for example, sniffles, pbsv and SVIM for long-read sequencing and Delly for short-read sequencing)[15,21]. We considered SVs that are consistent across two or more technologies. In total, we found 17,942 insertions, 15,419 deletions, 226 duplications, and 36 inversions in our high-confidence SV call set. Then, we compared the obtained multi-technology SV high-

confidence call set and the call set from our method. Our method's f1 score against the COLO829 multi-technology was 93.9% (precision: 96.0% and recall: 91.9%), where we excluded segmental duplications and complex repetitive regions, while the previous Dipasm-based SVs, Hifiasm+salsa2 and Hifiasm+3D-DNA f1 score was <82%, thereby demonstrating the major advantage of pstools method over existing methods in cancer genomics. To further confirm the pstools SV callset quality, we evaluated the SV calls for HG002 for which high-quality GIAB benchmark[22] is available, and we observed that the f1 score was 92.7% (precision: 94.2 % and recall: 91.3 %) (truvari -f ref.fasta -b base.vcf.gz−includebed HG002_SVs_Tier1_noVDJorXorY_v0.6.2.bed -o out−giabreport−passonly -r 1000 -p 0.00 -c c.vcf.gz) from our method.

Interestingly, our pstools method supports a homozygous 12 kbp deletion affecting PTEN on chromosome 10 (see Fig. 5). Our method can find SVs that occur due to a combination of multiple events on the same chromosome or different chromosomes producing breakage-fusion-bridge events; for example, chromosome 3 has fusion events from chromosomes 10, 12 and 6[15]. We also found a known breakage-fusion-bridge event on chromosome 15 that has insertions from chromosomes 6 and 20[15]. We also assessed the copy number profile from our SV calls against the copy number profile from the raw HiFi and Hi-C sequencing data. Fig 6 shows the coverage distribution of the raw HiFi and HiC data and the coverage distribution of the phased sequences, where we observe strong correlations between the sequencing technologies and our phased sequences. Overall, our datasets and chromosome-scale haplotype reconstruction pstools method provide a useful resource and streamlined approach for analysing the full spectrum of structural variations in complex cancer genomes that can potentially facilitate downstream haplotype-aware analyses of long-range promoter-enhancer interactions in regulatory networks. Thus, it provides a simple method for clinicians to dissect the full spectrum of SVs for individual patients that should facilitate better diagnosis and disease management, including predicting therapeutic responses.

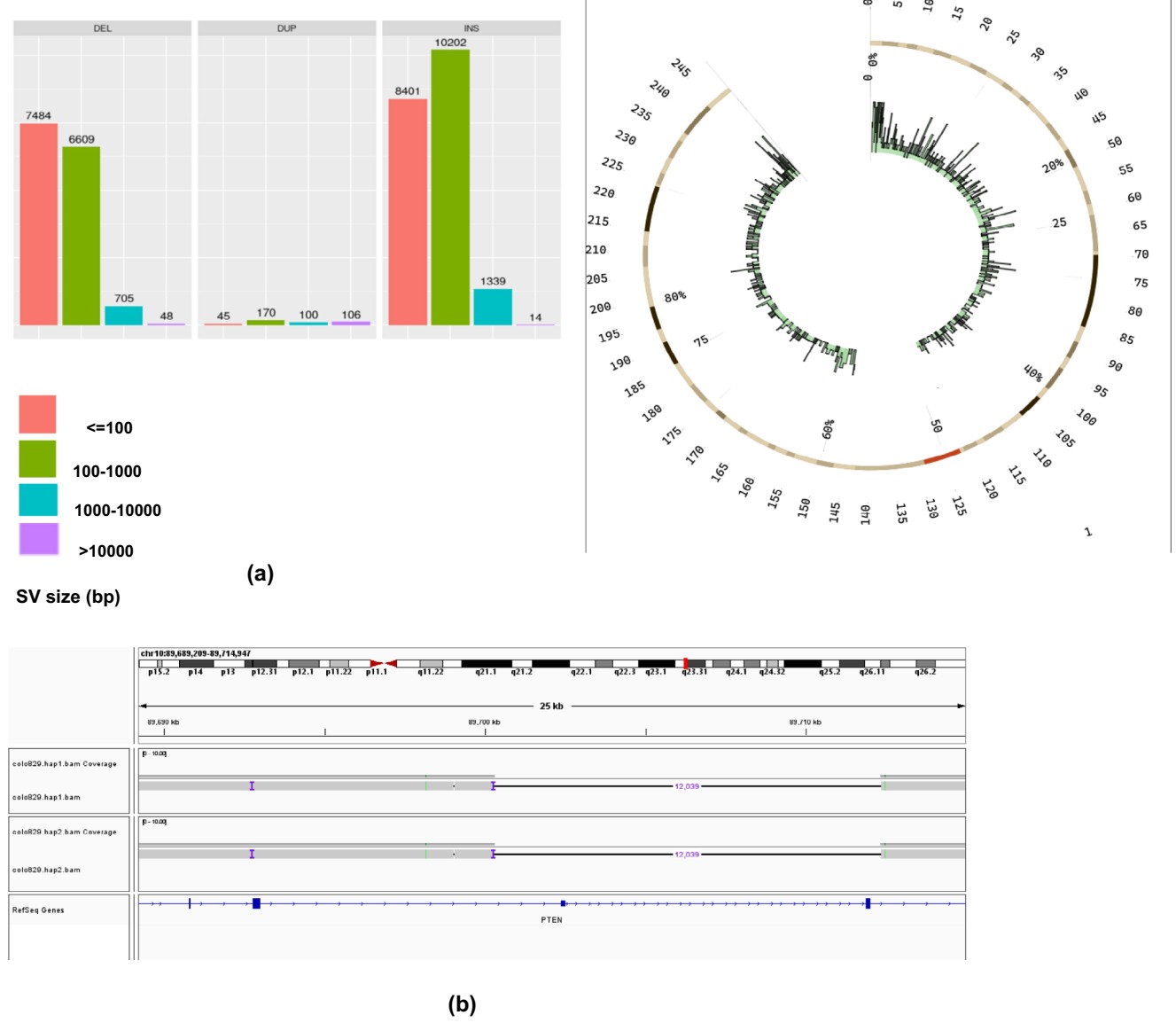

**Fig. 5 | a** Whole-genome precise SV characterisation of COLO829 (Top). SV types and size distributions and the circos plot shows SV distribution for chromosome 1. **b** Identification of a homozygous 12 kbp deletion affecting PTEN on chromosome 10 (Bottom). Source data are provided as a Source Data file.

## Discussion

Cancer genomes carry diverse and complex structural variations (SVs) ranging in size from 50 bp to whole chromosomes. We applied high-resolution sequencing techniques (HiFi and Hi-C) to the COLO829 cancer line and developed a fast and accurate tool (pstools) that combines local and global sequencing connectivity information from these data types to reconstruct chromosome-scale genomes useful for precise SV discovery. Our method produced chromosome-scale assemblies with a NG50 scaffold size of >130 Mb, switch/hamming error rates of <1.5% and an order of magnitude faster process (only <12 h), outperforming competing methods: trio-hifiasm and salsa2. Also, our method produced high-quality germline SV calls that were compared with the GIAB calls. In addition, we characterised somatic calls in repeat elements that were missed by short-read methods. However, our method will not be able to identify and characterise somatic genetic variation at low variant allele frequencies, suggesting the need to explore single-cell long-read approaches in future studies.

In our analysis, there were a few complex centromeric regions that were excluded from our phased sequences. The next potential step is to incorporate ultra-long (UL) nanopore sequencing data into the computational graphs (as already demonstrated for segmental duplication; https://doi.org/10.1101/2020.02.25.964445) to produce traversals for phased sequences in centromeres. Despite these limitations, our work enables the routine production of fully phased sequences at the chromosome scale that can be applied to hundreds/thousands of clinical and ethnically diverse samples for further biological discoveries.

## Methods

### Sample collection

We acquired human melanoma cell line COLO829 from ATCC (CRL-1974).

### DNA extraction

The frozen cell lines were briefly thawed and centrifuged for 2 min at $300 \times g$ to remove the supernatant containing cryopreservation media. The cell pellets were then washed with cold $1\times$ phosphate-buffered saline, centrifuged again, and the supernatant aspirated. High molecular weight genomic DNA (gDNA) was subsequently extracted from the cell pellets using the MagAttract HMW Kit (Qiagen) and following

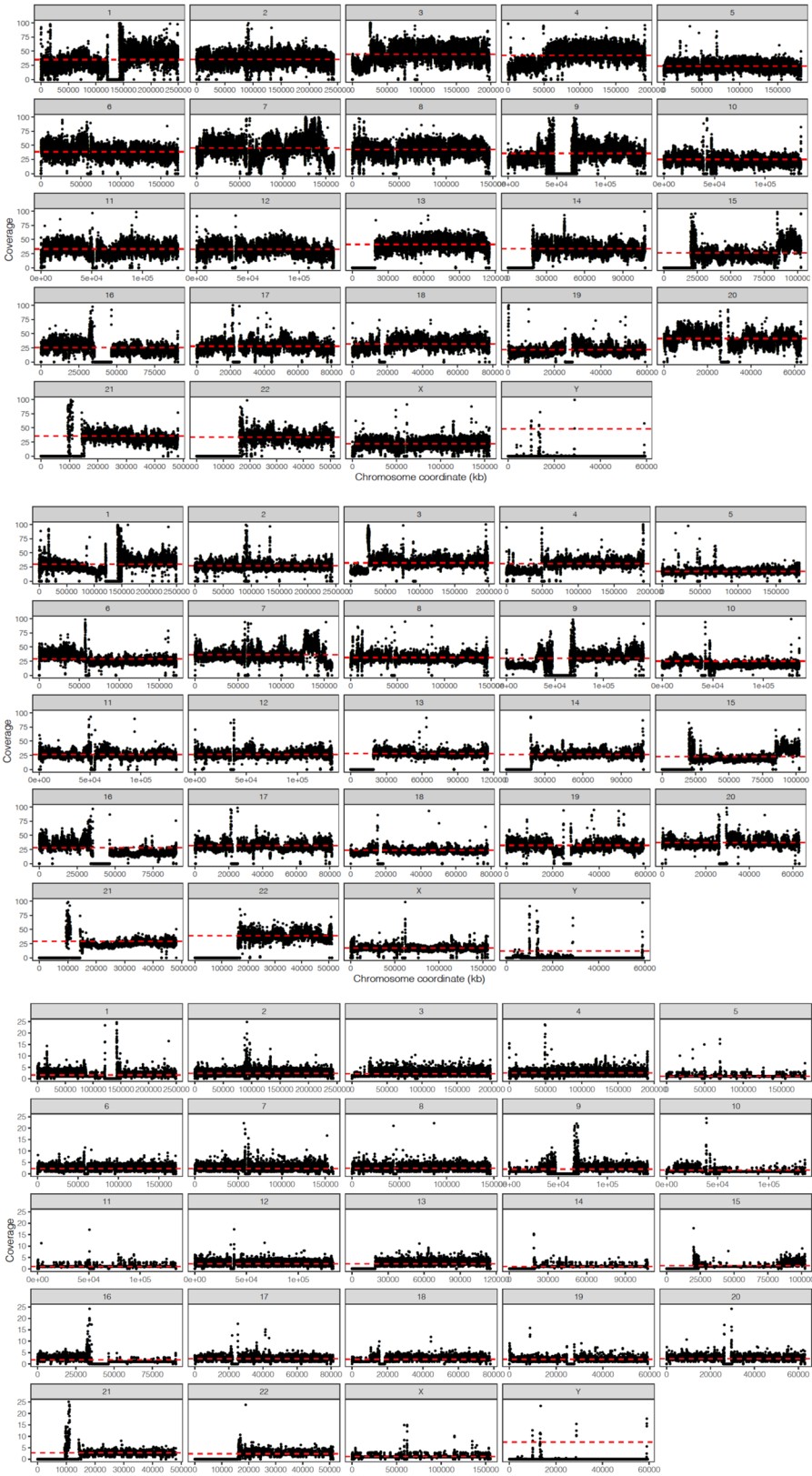

**Fig. 6 | Copy number profile correlation on all chromosomes.** Coverage distribution from HiFi (*Y* axis: 0–100) top and HiC data (*Y* axis: 0–100) middle and pstools phased sequences (*Y* axis: 0–25) bottom against reference genome, for visualisation of copy number profile correlation on all chromosomes (*X* axis: chromosome sizes). Source data are provided as a Source Data file.

the manufacturer's instructions. The extracted gDNA was then purified of excess buffers with 1× AMPure PB beads (Pacific Biosciences) before proceeding to library preparation.

## Library preparation and SMRT-sequencing

The purified gDNA was then prepped for PacBio single-molecule real-time (SMRT) sequencing using the Express Template Preparation Kit 2.0 (Pacific Biosciences) and following the manufacturer's instructions. Briefly, 6 µg of gDNA from each sample was sheared using Covaris g-TUBEs and then concentrated with 0.45× AMPure PB beads. The sheared gDNA was enzymatically treated to remove single-strand overhangs and repair nicked DNA templates, followed by an End Repair and A-tailing reaction to repair blunt ends and polyadenylate each template. Next, overhang SMRTbell adapters were ligated onto each template and purified using 0.45× AMPure PB beads to remove small fragments and excess reagents (Pacific Biosciences). The SMRTbell library was then treated with a cocktail of nucleases to remove damaged or unligated templates before subsequent size selection into 10 kb libraries using the BluePippin size fractionation system (Sage Biosciences). The final, size-selected library was then annealed to sequencing primer v2 and bound to sequencing polymerase 2.0 before being sequenced on multiple 8 M SMRTcells on the Sequel II system, each with a 30 h movie.

## HiFi sequencing

After data collection, the raw sequencing subreads were imported into the SMRTLink 9.0 bioinformatics tool suite (Pacific Biosciences) for processing. Intramolecular error correcting was performed using the circular consensus sequencing (CCS) algorithm to produce highly accurate (>Q20) CCS reads, each requiring a minimum of 3 polymerase passes.

## Hi-C sequencing

A Hi-C library was prepared using the Arima-HiC kit (P/N: A510008). The protocol uses 4 restriction enzymes to enable uniform per base coverage across the whole genome. This uniformity specifically helps in long-range analyses of variant discovery, base polishing, scaffolding, and phasing. Once the Armia-HiC protocol is applied, Illumina-compatible sequencing libraries were prepared that includes the process of shearing purified Arima-HiC ligation products and then size-selecting DNA fragments using SPRI beads. These fragments were enriched using Enrichment Beads provided in the Arima-HiC kit and converted into Illumina-compatible sequencing libraries using the Swift Accel-NGS 2S Plus kit (P/N: 21024) reagents. After adapter ligation, DNA was PCR amplified and purified using SPRI beads. The purified DNA underwent standard QC (qPCR and Bioanalyzer) and sequenced on the HiSeq X following the manufacturer's protocols.

## Pstools method

We first produced the Hifi sequence graph $G = (N,E)$ where $N = \{n1, n2...\}$ and $E = \{e1, e2, ...\}$ are a set of node sequences and edges respectively with standard tools (Hifiasm) hifiasm-r304 version. In the graph, we detected a set of bubbles $B = \{b1, b2...\}$ based on the state-of-the-art algorithm[23] and a set of alleles $A = \{a1, a2...\}$ in each bubble. On the graph, we mapped Hi-C read set $R = \{r1, r2...\}$ to the node sequences $N$ in the sequence graph $G$ using the k-mer approach ($k = 31$ for human genome). To accurately perform Hi-C mapping, we initially found the distinct k-mers between alleles in a bubble and then assigned the Hi-C reads to the correct allele $ak$ in a bubble $bm$ based on distinct k-mers. Thus, the linkage information from Hi-C read pairs on alleles in bubbles is stored within as well as across components in the graph. During Hi-C mapping, we also stored connectivity information on homozygous sequences in the graph. There are <20% reads out of total mapped to the heterozygous bubbles (hets), while remaining to homozygous, and combinations. The sensitivity and specificity of read mapping is >95% compared to short-read alignment that is followed by traditional scaffolders, often very computationally expensive (>15 h). While our method performed the mapping step in <2.5 h. The goal of pstools is to find a set of haplotype paths $H = \{n1, n2....\}$ maximum supported by reads $R = \{r1, r2...\}$ within and across components on the chromosome-scale. We performed a phased scaffolding process that consists of two steps: first, we found haplotype paths through the chain of bubbles for every component based on Hi-C read support (>5 reads); second, we combined the phasing and scaffolding information across components to produce chromosome-scale haplotype paths.

In the first step, we connected alleles in bubble chains from Hi-C reads support in a greedy manner based on maximum support of reads (phased contigs). In the second step, the goal is to connect haplotypes across components. We formulate this problem as a graph problem. We represent phased contigs and Hi-C contact information in a phased scaffold graph, where nodes represent the haplotype in a specific orientation from components and edges represent the Hi-C reads support. We additionally calculate the edge confidence score ($e/(i + j)$, where $e$ is a hic coverage, $i$ and $j$ are the sum of coverages of adjacent edges of respective nodes of a given edge) on how well a given edge and its neighbours are supported by Hi-C reads (neighbourhood property). Finding phased scaffolds through this graph is the k-partite matching problem, where k is the 2 * #chromosomes. Since k-partite matching is a NP hard problem, we solve using greedy heuristics. In the greedy procedure, we first start with longer node sequences >10 Mb and greedily make k-partitions such that neighbourhood property is preserved in each partition. After these iterations finish, we include the remaining shorter contigs to partitions under the condition that the p-neighbouring contigs (where $p = 3$) are well supported in every partition, resulting in final partitions. Interestingly, the order and orientation of haplotype contigs are generally implied from the neighbourhood property within the partitions. We performed one additional round of ordering and orientation check and correction based on neighbourhood property. This procedure produced partitions representing scaffolds for one haplotype and the corresponding partitions for other haplotype are constructed by considering the corresponding haplotype contigs for every partition for diploid case. For polyploidy cases, iterate the greedy procedure again for remaining haplotype contigs that require further exploration. Finally, the phased scaffold sequences are spelled out from these phased partitions consisting of ordered contigs with proper orientations. The algorithm is ploidy agnostic in principle. The overall algorithm performance for the complex ploidies depends on Hi-C data quality from homologue spatial positioning in 3D space.

## Hi-C based phasing evaluation

For local phasing and global phasing evaluation, we computed the switch and hamming error rates respectively. The switch error rate is the number of local switches divided by #heterozygous sites, while the hamming error rate is the hamming distance on the global level divided by #heterozygous sites. We implemented an efficient k-mer based method and used maximum Hi-C read support to detect switch errors on heterozygous positions. The process is to first find heterozygous k-mers (hets) from phased assemblies using 31-mers. After that, we map Hi-C reads to the assemblies using 31-mers. If there are >5 reads that support a switch between consecutive hets in assemblies, we consider a haplotype switch. In the hamming error calculations, we consider every switch support het pair in hamming distance for a global view of phasing errors (this implicitly penalises any long switches). We perform this operation for the whole scaffold/contigs over all scaffolds/contigs. This evaluation operation is made available in pstools subcommand

phasing_error. Even Hi-C based pstools phasing evaluation has been experimented with and applied to diverse HPRC samples. (https://doi.org/10.1101/2022.07.09.499321).

**Reporting summary**

Further information on research design is available in the Nature Portfolio Reporting Summary linked to this article.

## Data availability

The COLO829 datasets are made available at SRA accessions: SRR22761284 for HiFi and SRR22761283 for Hi-C sequencing. Previously published datasets: HG002 Hifi and Hi-C are publicly available from the HPRC project at https://s3-us-west-2.amazonaws.com/human-pangenomics/index.html?prefix=working/HPRC_PLUS/HG002/raw_data/hic/downsampled/. HG002 Illumina is publicly available at https://s3-us-west-2.amazonaws.com/human-pangenomics/index.html?prefix=NHGRI_UCSC_panel/HG002/hpp_HG002_NA24385_son_v1/ILMN/downsampled/. Source data are provided with this paper.

## Code availability

The codebase is publicly available at https://github.com/shilpagarg/pstools.git and docker/biocontainer version: quay.io/biocontainers/pstools:0.2a3−hd03093a_1.

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

## Acknowledgements

The author is thankful to George Church from Harvard Medical School, Anthony Schmitt from Arima Genomics and Nancy Francoeur for Mt. Sinai for providing necessary support on sequencing technologies. Thanks are also due to the Human Pangenome Reference consortium (HPRC) for HiFi and Hi-C datasets. Lastly, the author acknowledges the productive comments from anonymous reviewers and the support from the Novo Nordisk Foundation (NNF21OC0069089).

## Author contributions

S.G. conceived the project, implemented it, and drafted the manuscript.

## Competing interests

The author declares no competing interests.
