## [Peer review file · Nature Communications]

REVIEWER COMMENTS

Reviewer #2 (Remarks to the Author):

I thank the author for their response. Unfortunately, most of my original request are still unanswered either in part or completely. I am using the original numbering below:

1. A short description of the method and a reference should be added to the manuscript text - I was not able to find it in the revised version.
2. The author's response implies that NGA50 was computed, but I was not able to find the numbers anywhere in the text. The new figure 4 shows the chromosomal mappings of the pstools assemblies, but it is not a substitute of the NGA50 evaluation. My request is to report the NGA50 number for all assemblies, using the authors method of choice.
3. It is still unclear which table entries represent contigs, and which represent scaffolds. This should be clearly labelled.
4. I was not able to find the separate statistics for precision and recall. The authors' response should also be added to the paper text. Finally, all other COLO829 assemblies should be evaluated using benchmark SV set, not just pstools.
5. I disagree with the authors' reasoning to dismiss my original request. pstools assemblies are already compared against other assemblies (e.g. in Table 1) using various other assembly metric. SV recall/precision on the GIAB curated SV set could be easily computed (similarly to how it's done for pstools assemblies) and should be reported.
7. Pre-compiled binary does not guarantee reproducibility on a variety of OS and computational environments. I am asking to provide a container package (bioconda/docker/singularity etc) to ensure that the software will be easily available to the users. This is standard practice for most of the bioinformatics tools published in similar journals.

Reviewer #3 (Remarks to the Author):

While this study has improved I have a remaining important concern, ie I am somewhat concerned that the author may still overcall somatic SVs (i.e. assign actual germline SVs a “somatic SV status” erroneously). Can the author please greedily subtract germline SVs by filtering out known germline SV loci from recent long-read sequencing based studies (HGSVC and HPRC) from their colon cancer cell line SV callset? What fraction of their “somatic SV” calls will be removed if such a filter (e.g. using a 50% reciprocal overlap to filter germline SVs) is used? This number should be discussed in the paper, as one would clearly expect that only a subset of the SVs identified in the colon cancer cell line are somatic – and it would be important to ensure there is clarity on this. With studies such as the HGSVC (and to some extent also HPRC) having looked rather extensively at germline SVs in the past 2 years, it should be rather straightforward to develop a simple, reasonable filter to identify likely germline variants in the colon cancer cell line (i.e. to avoid overcalling somatic SVs).

To clarify, I would expect a title such as "Towards routine chromosome-scale haplotype-resolved reconstruction in cancer genomics" to imply that somatic SVs are most carefully filtered from germline SVs.

Reviewer #4 (Remarks to the Author): Expert in genome informatics, haplotype assembly, and long-read sequencing

Pstools

I have been asked to provide comments on the responses to Reviewer #1. In overall, I find that only part of what Reviewer #1 (and the other reviewers) asked have been addressed, and some requires further revision.

1. Are the NG50 results in Table 1 Contig NG50s? The first paragraph of the Benchmarking section vaguely distinguishes contig vs. scaffold NG50s, and lumps them into “assembly size NG50”. Please clarify what has been measured. Scaffolding methods are designed to produce chromosome-scale “scaffolds”, not “contigs”, thus comparing contig NG50s to pstoool may not be a side-by-side comparison.

If the NG50s are measuring scaffold NG50s for scaffolded assemblies, it should be stated as so. The discussion states “Our method produced chromosome-scale assemblies with a NG50 scaffold size of >130 Mb”, which made me more confusing. Also, it wasn’t clear to me how pstoools was connecting chromosomal arms such as demonstrated for chr1 and chr6 in Fig. 3. How were the disconnected components linked? Are there gaps placed in? If so, I wonder what the NG50 is reported for pstoools. Perhaps reporting both contig and scaffold NG50 in Table 1 will resolve all these concerns.

2. How is the repetitive sequence resolved in Fig. 3, left panel? Will the sequence going to be present in both chr13 and chr14? Which repeat is that? I would be surprised if that is the rDNA, and it’s completely identical between the two chromosomes. How does the author confirm the structural accuracy and the correct representation of the repeat that has been resolved?

3. The GIAB benchmark is only available for HG002. Please clarify which results are reported from HG002. The result section of the Benchmarking begins with results from all 3 human genome assemblies, however suddenly switches into results only applicable to HG002.

4. Indicate which hamming / switch errors were generated specifically by Yak / GIAB / pstoools with superscripts on each numbers, i.e. place “a” next to the numbers produced with GIAB and provide details in legend such as “a, GIAB”. This will clear the confusion raised by both Reviewer #1 and #2.

5. The ideogram of phased sequences (Fig. 4) shows rough alignment blocks of sequences from HG002. I would expect two blocks for each chromosome at the same locus though, for each phase, which would differ in length at some locations. Also, what are the gaps in between? Distinguish the gaps present in GRCh38, scaffolding gaps in pstoools, and mis-joins / missed-joins identified via rapid-curation. Perhaps T2T-CHM13 would be a better reference to use, which will be able to show the phased sequences extending to gaps in GRCh38.

6. All the results are substantially missing supporting evidence for structural accuracy. At minimum, show the same type of ideogram with contents pointed in Fig 4 along with the rapid-curation results for all other human samples in Supplementary figures and/or tables. The only way to infer the structural accuracy of pstoools is by inferring it from the human genome assemblies, as there is no true linear genome assembly available for the COLO829 cancer cell line. Otherwise, to claim that pstoools works on cancer genomes with high level of aneuploidy and structural re-arrangements, creating a synthetic cancer genome using simulated HiFi and Hi-C reads may be a classic way to validate the structural accuracy of the assemblies generated by pstoools, however I wouldn’t recommend this as the read simulations wouldn’t always reflect the true error rates / interaction nature in real sequencing reads.

7. In line with this, provide more details of the COLO829 assembly SV call results. I was expecting more precise numbered results to see for the variant calls equivalent to true positive / false negative / false positive. The 4-way comparison in Fig. 1b is somewhat useful, however the numerous colors makes difficult to distinguish which calls are considered as FN or FP. In addition, the figure only shows SV calls in repeat elements. Are there no SVs called outside of the repeat regions? How are the SVs involving multiple / composite repeats handled? I’d suggest providing an additional Supp. Table with the numbers, and grouping those to TP / FN / FP.

8. Move the example of somatic and germline SV calling results in Introduction under the cancer cell line result section, before Fig. 5. Those aren’t “examples”, they are “results”.

9. What is the context that makes the 12kb deletion in PTEN “interesting”? Fig. 5b isn’t showing the relative context of the deletion and the PTEN gene. The legend is also poorly written. What is shown in the IGV screen-shot? I see only one alignment with a deletion, presumably from the pstoools assembly. Where is the other haplotype(s)? Is there only one contig assembled covering this region? Please provide more insights of this deletion, relative to what has been already known or described before.
10. How different is the size distribution compared to the original SV call set from Enrique Velazquez-Villarreal et al. 2020? Are there any patterns of the variants / re-arrangements newly characterized?
11. The circus plot in Fig. 5a (right) is still missing color legends of the most outer circle (ideogram). At minimum, highlight the two translocations mentioned in the main text. It is difficult to track which ones are the translocations the author is mentioning. Also, insertions and deletions are not so informative as it seems like happening everywhere in the genome. Is this what the author wants to present?
12. Cite the relevant paper for the breakage-fusion-bridge event where it is mentioned in the text where it is mentioned.
13. Copy number profile in Fig. 6 shows recurrent copy gains and losses across all chromosomes in HiFi / Hi-C / pstoools contigs. However, it is a bit difficult to track the relative differences in each chromosome. I’d suggest to 1) adjust the Y axis on the top and middle panels (HiFi and Hi-C) as estimated copy numbers or relative copy numbers to show gain / losses, 2) draw estimated copy-number lines for each segments, 3) color the lines by HiFi / Hi-C / pstoools assembly, and 4) overlay them in one panel for each chromosome. This way, the relative correlation will be seen more clearly. Currently, for example, the copy gain in the q-arm (or loss in p-arm?) in chr4 from pstoools looks not as pronounced as in the HiFi and Hi-C reads due to the scale difference.
14. Methods for evaluation is largely missing, or placed in the Results section.

Below are a few minor comments, some related to the newly added text during revision.

1. Until which part is the introduction? The paragraph beginning with “For the first time, ...” on the 2nd page seems to be the last paragraph of the Introduction, however the newly added text during revision makes it part of the Results section. Please provide proper section headers.
2. In “However, the high-resolution methods are scanty ...” in page 1, “re-sequencing methods” is a more widely used instead of “high-resolution methods”. On the same page, in “... limits its applications because trios are not readily available.”, consider rephrase “readily” to “routinely” or “always”.
3. Fig. 1b is missing SC only
4. What’s the difference between Fig. 1b and 1c? Text says “Fig 1b presents the distribution of germline SV calls ... than somatic calls in Fig 1c”. Legends in both panels indicate somatic SVs. Please correct.
5. Rephrase the text in discussion: “One limitation is that our method will not be able to identifying and characterizing somatic genetic variation ...” to “One limitation is that our method will not be able to identify and characterize somatic variation...”.

REVIEWER COMMENTS

Reviewer #2 (Remarks to the Author):

I thank the author for their response. Unfortunately, most of my original request are still unanswered either in part or completely. I am using the original numbering below:

1. A short description of the method and a reference should be added to the manuscript text - I was not able to find it in the revised version.

We have now added a paragraph in the method section: “Hi-C based phasing evaluation. For local phasing and global phasing evaluation, we computed the switch and hamming error rates respectively. The switch error rate is the number of local switches divided by #heterozygous sites, while the hamming error rate is the hamming distance on the global level divided by #heterozygous sites. We implemented an efficient k-mer based method and used maximum Hi-C read support to detect switch errors on heterozygous positions. The process is to first find heterozygous k-mers (hets) from phased assemblies using 31-mers. After that, we map Hi-C reads to the assemblies using 31-mers. If there are >5 reads that support a switch between consecutive hets in assemblies, we consider a haplotype switch. In the hamming error calculations, we consider every switch support het pair in hamming distance for a global view of phasing errors (this implicitly penalizes any long switches). We perform this operation for the whole scaffold/contigs over all scaffolds/contigs. This evaluation operation is made available in pstools subcommand phasing_error. Even Hi-C based pstools phasing evaluation has been experimented and applied to diverse HPRC samples. (<https://doi.org/10.1101/2022.07.09.499321>).”

2. The author's response implies that NGA50 was computed, but I was not able to find the numbers anywhere in the text. The new figure 4 shows the chromosomal mappings of the pstools assemblies, but it is not a substitute of the NGA50 evaluation. My request is to report the NGA50 number for all assemblies, using the authors method of choice.

We have now added a column on NGA50 calculations in Table 1.

Dataset	Method	Total size	NG50	NGA50	QV	Hamming/Switch
HG002	pstools ^s	6.13 Gb	132.3 Mb	81.9 Mb	51.6	1.2/0.4 ^b
	trio-hifiasm ^c	5.94 Gb	78.9 Mb	64.1 Mb	51.5	1.6/0.8 ^b
	Hifiasm (hi-c) ^c	5.92 Gb	53.7 Mb	33.7 Mb	52.0	0.9/0.9 ^b
	Dipasm	5.93	24.7 Mb	13.9 Mb	41.2	0.5/0.5 ^b

	(3D-DNA) ^C	Gb				
	pstools+sals a2 ^S	6.13 Gb	125.5 Mb	70.7 Mb	51.5	49.5/0.72^b
	trio-hifiasm + salsa2 ^S	5.96 Gb	137 Mb	70.9 Mb	51.5	1.1/0.6^b
HG0073 3	pstools^S	6.21 Gb	131.9 Mb	70.7 Mb	51.1	0.6/0.2^a
	trio-hifiasm ^C	6.17 Gb	43 Mb	55.3 Mb	51.1	1.2/1.8 ^a
	Hifiasm (hi-c) ^C	6.06 Gb	42.5 Mb	29.9 Mb	50.0	1.3/1.0 ^a
	Dipasm ^C (3D-DNA)	5.93 Gb	25.7 Mb	16.3 Mb	41.3	1.2/0.3^a
	pstools+sals a2 ^S	6.17 Gb	155 Mb	70.5 Mb	51.1	48.5/0.52^a
	trio-hifiasm + salsa2 ^S	6.07 Gb	135.5 Mb	70.4 Mb	51.1	1.1/0.9^a
PGP1	pstools^S	6.14 Gb	111.7 Mb		-	1.2/0.9 ^c
	pstools+sals a2 ^S	6.13 Gb	120.5 Mb	69.7 Mb	-	48.5/0.8 ^c
	Dipasm ^C (3D-DNA)	5.94 Gb	16.5 Mb	10.65 Mb	-	1.7/0.3 ^c
COLO82 9	pstools^S	6.13 Gb	132.1 Mb	54.8 Mb	-	1.3/0.6^c
	pstools+sals a2 ^S	6.12 Gb	100.1 Mb	54.7 Mb	-	47.5/0.3^c
	Hifiasm+sals a2 ^S	6.03 Gb	1.7 Mb	0.6 Mb	-	-
	Hifiasm+3D- DNA ^S	6.05 Gb	2.5 Mb	1.2 Mb	-	-
	Dipasm ^C (3D-DNA)	5.92 Gb	12.4 Mb	5.6 Mb	-	20.5/5.6^c

3. It is still unclear which table entries represent contigs, and which represent scaffolds. This should be clearly labelled.

Thank you for your suggestion. We have now added superscripts in Table 1 to represent contigs (C) and scaffolds (S).

4. I was not able to find the separate statistics for precision and recall. The authors' response should also be added to the paper text. Finally, all other COLO829 assemblies should be evaluated using benchmark SV set, not just pstools.

We have now added precision and recall to the paper text. Also added evaluation for all assemblies of COLO829.

5. I disagree with the authors' reasoning to dismiss my original request. pstools assemblies are already compared against other assemblies (e.g. in Table 1) using various other assembly metric. SV recall/precision on the GIAB curated SV set could be easily computed (similarly to how it's done for pstools assemblies) and should be reported.

The focus of the manuscript is for methods for cancer genomes. Therefore, we aimed to present SV precision and recall for COLO829 (as requested above) from all assemblies only and avoided diluting the focus of the manuscript.

7. Pre-compiled binary does not guarantee reproducibility on a variety of OS and computational environments. I am asking to provide a container package (bioconda/docker/singularity etc) to ensure that the software will be easily available to the users. This is standard practice for most of the bioinformatics tools published in similar journals.

Thank you for your suggestion. We have made it available in docker/biocontainer:

`"quay.io/biocontainers/pstools:0.2a3--hd03093a_1"`.

Reviewer #3 (Remarks to the Author):

While this study has improved I have a remaining important concern, ie I am somewhat concerned that the author may still overcall somatic SVs (i.e. assign actual germline SVs a "somatic SV status" erroneously). Can the author please greedily subtract germline SVs by filtering out known germline SV loci from recent long-read sequencing based studies (HGSVC and HPRC) from their colon cancer cell line SV callset? What fraction of their "somatic SV" calls will be removed if such a filter (e.g. using a 50% reciprocal overlap to filter germline SVs) is used? This number should be discussed in the paper, as one would clearly expect that only a subset of the SVs identified in the colon cancer cell line are somatic – and it would be important to ensure there is clarity on this. With studies such as the HGSVC (and to some extent also HPRC) having looked rather extensively at germline SVs in the past 2 years, it should be rather straightforward to develop a simple, reasonable

filter to identify likely germline variants in the colon cancer cell line (i.e. to avoid overcalling somatic SVs).

Thank you for your great suggestion. We have now carefully avoided overcalling and revised suitably: "For example, we applied our graph-based method for somatic and germline calling for COLO829, where somatic calls were produced by subtracting the germline calls for COLO829BL^{17,18} and HG002 GIAB benchmark (https://ftp-trace.ncbi.nlm.nih.gov/ReferenceSamples/giab/release/AshkenazimTrio/HG002_NA24385_son/latest/GRCh37/). As a result, we found 166 somatic structural variant calls at precise base-level resolution on the whole genome. We even specifically characterized these calls in complex repeat elements including ALUs, L1, L2, and benchmarked against publicly available calls produced using multiple technologies/tools, for example, a short-read-based NYGC call set (<https://www.nygenome.org/bioinformatics/3-cancer-cell-lines-on-2-sequencers/>), single-cell-based call set (available from Enrique Velazquez-Villarreal et al. 2020)²² and multi-technology-based UMCU call set (https://github.com/UMCUGenetics/COLO829_somaticSV). Figure 1b presents the 4-way comparison of these

callsets, where each bar represents the agreement of callsets in repeat elements. We observed that our calls agreed with the single-cell benchmarked callset (blue) and UMCU (yellow), but our method provides more precise SV characterization at the base and haplotype resolution. The green and red and pink colored bar represent calls missed by pstools possibly due to low variant allele frequency but found in single-cell method. Additionally, we produced higher calls in comparison to bulk short-read-based calls from NYGC. Out of these somatic variants, we found 15 novel calls and 6 calls fall in the COSMIC genes. Fig 1c presents the distribution of germline SV calls in repeat elements, indicating the higher rate of germline calls than somatic calls in Fig 1b. From above, it follows that our pstools method can produce precise and comprehensive haplotype-aware SV landscapes over short-read sequencing technologies in repetitive regions.”.

And revised Figure 1 suitably.

To clarify, I would expect a title such as "Towards routine chromosome-scale haplotype-resolved reconstruction in cancer genomics" to imply that somatic SVs are most carefully filtered from germline SVs.

Considering word limit and scope, we considered to retain the same title.

Reviewer #4 (Remarks to the Author): Expert in genome informatics, haplotype assembly, and long-read sequencing

Pstools

I have been asked to provide comments on the responses to Reviewer #1. In overall, I find that only part of what Reviewer #1 (and the other reviewers) asked have been addressed, and some requires further revision.

Thank you for your suggestion on points for further revision. We have now significantly addressed your points, however, there are some points that require extensive work that will serve as a material for another manuscript. For example, preparing somatic SV benchmarks for melanoma like GIAB HG002 benchmarks by comparing across multiple methods require extensive community efforts similar to GIAB for cancer genomics. Nonetheless, we have presented necessary benchmarking and beyond. Also, this study presents cutting-edge

sequencing technologies applied for the first time to cancer genomics as well as a novel computational method to construct cancer genomes useful for SV calling, thus setting a milestone in cancer genomics.

1. Are the NG50 results in Table 1 Contig NG50s? The first paragraph of the Benchmarking section vaguely distinguishes contig vs. scaffold NG50s, and lumps them into “assembly size NG50”. Please clarify what has been measured. Scaffolding methods are designed to produce chromosome-scale “scaffolds”, not “contigs”, thus comparing contig NG50s to pstool may not be a side-by-side comparison. If the NG50s are measuring scaffold NG50s for scaffolded assemblies, it should be stated as so. The discussion states “Our method produced chromosome-scale assemblies with a NG50 scaffold size of >130 Mb”, which made me more confusing. Also, it wasn’t clear to me how pstools was connecting chromosomal arms such as demonstrated for chr1 and chr6 in Fig. 3. How were the disconnected components linked? Are there gaps placed in? If so, I wonder what the NG50 is reported for pstools. Perhaps reporting both contig and scaffold NG50 in Table 1 will resolve all these concerns.

Thank you for the good suggestion. We have now added superscripts in column 1 to indicate contigs and scaffolds depending on the method used.

Also, the text is revised suitably: “1) it correctly disentangles the chromosomes (not resolve repeats, but connect components using long-range Hi-C with 100 N’s)” and the caption of Figure 4: “ Ideogram of phased sequences of HG002 (left) and COLO829 (right) - One color for each chromosome representing chromosome level. No color represent gaps for complex regions, for example, centromeric, acrocentric, etc.”

2. How is the repetitive sequence resolved in Fig. 3, left panel? Will the sequence going to be present in both chr13 and chr14? Which repeat is that? I would be surprised if that is the rDNA, and it’s completely identical between the two chromosomes. How does the author confirm the structural accuracy and the correct representation of the repeat that has been resolved?

The Hi-C is not able to resolve through repeats, though helps in long-range connections. The structural accuracy is confirmed by ideogram in Fig 4 against the reference sequence and benchmarking using standard evaluation metrics and tools (NGA50 and <https://gitlab.com/wtsi-grit/rapid-curation>).

3. The GIAB benchmark is only available for HG002. Please clarify which results are reported from HG002. The result section of the Benchmarking begins with results from all 3 human genome assemblies, however suddenly switches into results only applicable to HG002.

We added the suggested information in the text: “a phasing accuracy of >98.5% against Genome in a Bottle (GIAB) variant calls (see Table 1) for HG002 as per availability”.

4. Indicate which hamming / switch errors were generated specifically by Yak / GIAB / pstools with superscripts on each numbers, i.e. place “a” next to the numbers produced with GIAB and provide details in legend such as “a, GIAB”. This will clear the confusion raised by both Reviewer #1 and #2.

Done

5. The ideogram of phased sequences (Fig. 4) shows rough alignment blocks of sequences from HG002. I would expect two blocks for each chromosome at the same locus though, for each phase, which would differ in length at some locations. Also, what are the gaps in between? Distinguish the gaps present in GRCh38, scaffolding gaps in pstools, and mis-joins / missed-joins identified via rapid-curation. Perhaps T2T-CHM13 would be a better reference to use, which will be able to show the phased sequences extending to gaps in GRCh38.

We agree that T2T-CHM13 is a better reference, however, the focus of the manuscript is not on centromeres. There are no assembled sequences (alignments) in centromeres and acrocentromeres (chr13, chr14, chr15, chr21, chr22) forming gaps shown in Fig 4.

6. All the results are substantially missing supporting evidence for structural accuracy. At minimum, show the same type of ideogram with contents pointed in Fig 4 along with the rapid-curation results for all other human samples in Supplementary figures and/or tables. The only way to infer the structural accuracy of pstools is by inferring it from the human genome assemblies, as there is no true linear genome assembly available for the COLO829 cancer cell line. Otherwise, to claim that pstools works on cancer genomes with high level of aneuploidy and structural re-arrangements, creating a synthetic cancer genome using simulated HiFi and Hi-C reads may be a classic way to validate the structural accuracy of the assemblies generated by pstools, however I wouldn't recommend this as the read simulations wouldn't always reflect the true error rates / interaction nature in real sequencing reads.

We have now added these simulations and revised the text suitably: "To further assess the structural accuracy of COLO829, we applied WHdenovo (whdenovo simulate: <https://github.com/shilpagarg/WHdenovo/tree/master/whdenovo>) to simulate haplotypes by incorporating COLO829 ground truth SNP and SV mutations in the Grch37 human genome. The HiFi and Hi-C read simulations are performed at 40x fold using pbsim and wgsim respectively. We then applied pstools on these data types that produced assemblies with a NG50 scaffold size of 131.5 Mb. We also additionally compared predicted SV calls against the ground truth and the resultant precision and recall are 97.1% and 95.8% respectively. These results suggest that the pstools produces high-quality assemblies and SV calls. Thus these simulation experiments further confirm the structural accuracy of pstools assemblies."

Additionally, we presented ideogram for COLO829.

7. In line with this, provide more details of the COLO829 assembly SV call results. I was expecting more precise numbered results to see for the variant calls equivalent to true positive / false negative / false positive. The 4-way comparison in Fig. 1b is somewhat useful, however the numerous colors makes difficult to

distinguish which calls are considered as FN or FP. In addition, the figure only shows SV calls in repeat elements. Are there no SVs called outside of the repeat regions? How are the SVs involving multiple / composite repeats handled? I'd suggest providing an additional Supp. Table with the numbers, and grouping those to TP / FN / FP.

We have added true values of precision and recall.

We now performed careful filtering in Fig 1b and also reported numbers for the whole genome. Yes, we observed cases of SVs spanning multiple repeats, so they were considered in all.

Extensive benchmarking across methods (like a GIAB benchmarking effort) is beyond the scope of this manuscript at this stage that is planned as a follow-up study.

8. Move the example of somatic and germline SV calling results in Introduction under the cancer cell line result section, before Fig. 5. Those aren't "examples", they are "results".

The subheading "results" is now added.

9. What is the context that makes the 12kb deletion in PTEN "interesting"? Fig. 5b isn't showing the relative context of the deletion and the PTEN gene. The legend is also poorly written. What is shown in the IGV screen-shot? I see only one alignment with a deletion, presumably from the pstools assembly. Where is the other haplotype(s)? Is there only one contig assembled covering this region? Please provide more insights of this deletion, relative to what has been already known or described before.

Thank you for the suggestions.

We now showed both haplotypes.

However, the impact of PTEN, for example, in gene expression and functional level on the long-range context, is beyond the scope of this manuscript.

10. How different is the size distribution compared to the original SV call set from Enrique Velazquez-Villarreal et al. 2020? Are there any patterns of the variants / re-arrangements newly characterized?

Our method can now characterize precise patterns at the base-level and haplotype resolution. 15 calls are novel specifically in the repetitive elements.

11. The circus plot in Fig. 5a (right) is still missing color legends of the most outer circle (ideogram). At minimum, highlight the two translocations mentioned in the main text. It is difficult to track which ones are the translocations the author is mentioning. Also, insertions and deletions are not so informative as it seems like happening everywhere in the genome. Is this what the author wants to present?

For better visualisation, we now presented the SV distribution on chromosome 1 only (also similar to presentations used in our previous Dipasm article).

12. Cite the relevant paper for the breakage-fusion-bridge event where it is mentioned in the text where it is mentioned.

Done.

13. Copy number profile in Fig. 6 shows recurrent copy gains and losses across all chromosomes in HiFi / Hi-C / pstools contigs. However, it is a bit difficult to track the relative differences in each chromosome. I'd suggest to 1) adjust the Y axis on the top and middle panels (HiFi and Hi-C) as estimated copy numbers or relative copy numbers to show gain / losses, 2) draw estimated copy-number lines for each segments, 3) color the lines by HiFi / Hi-C / pstools assembly, and 4) overlay them in one panel for each chromosome. This way, the relative correlation will be seen more clearly. Currently, for example, the copy gain in the q-arm (or loss in p-arm?) in chr4 from pstools looks not as pronounced as in the HiFi and Hi-C reads due to the scale difference.

The current plot provides a clear copy number estimates from the raw sequencing data and assemblies. However, there are no tools specifically designed to estimate the copy number profile from HiFi, Hi-C and assemblies at this stage. Therefore, to avoid any bias of previous tools, we would like to present the copy number profile from the coverage distribution from sequencing data on the whole genome. The x-axis is the same in all plots, the y-axis represents the coverage. Although the scale of the y-axis is different, the overlapping reads (30x) are merged into a consensus sequence, therefore, we observe one-on-one relations between reads and assemblies. Moreover, the relative distribution between them looks similar in the plot.

14. Methods for evaluation is largely missing, or placed in the Results section.

We have added the necessary explanation for evaluation methods.

Below are a few minor comments, some related to the newly added text during revision.

1. Until which part is the introduction? The paragraph beginning with “For the first time, ...” on the 2nd page seems to be the last paragraph of the Introduction, however the newly added text during revision makes it part of the Results section. Please provide proper section headers.

Done

In “However, the high-resolution methods are scanty ...” in page 1, “re-sequencing methods” is a more widely used instead of “high-resolution methods”. On the same page, in “... limits its applications because trios are not readily available.”, consider rephrase “readily” to “routinely” or “always”.

Done

3. Fig. 1b is missing SC only

Done

4. What’s the difference between Fig. 1b and 1c? Text says “Fig 1b presents the distribution of germline SV calls ... than somatic calls in Fig 1c”. Legends in both panels indicate somatic SVs. Please correct.

Done

5. Rephrase the text in discussion: “One limitation is that our method will not be able to identifying and characterizing somatic genetic variation ...” to “One limitation is that our method will not be able to identify and characterize somatic variation...”.

Done

REVIEWERS' COMMENTS

Reviewer #2 (Remarks to the Author):

The author has addressed all my concerns, I have no further requests.

Reviewer #3 (Remarks to the Author):

The author has addressed and resolved my outstanding comments in a satisfactory manner

Reviewer #4 (Remarks to the Author):

All my comments and concerns raised have been adequately addressed. The manuscript has been substantially improved.

REVIEWERS' COMMENTS

Reviewer #2 (Remarks to the Author):

The author has addressed all my concerns, I have no further requests.

Reviewer #3 (Remarks to the Author):

The author has addressed and resolved my outstanding comments in a satisfactory manner

Reviewer #4 (Remarks to the Author):

All my comments and concerns raised have been adequately addressed. The manuscript has been substantially improved.

All the reviewers are glad that the authors have successfully addressed all their comments to their satisfaction. Therefore, no further response is expected from the authors.